# Factors leading to the late diagnosis and poor outcomes of breast cancer in Matabeleland South and the Bulawayo Metropolitan Provinces in Zimbabwe

**Munyaradzi S. Magara** [1]*, **Simbarashe G. Mungazi**[1], **Peeps Gonde**[2], **Hausitoe Nare**[3], **Desmond Mwembe**[3], **Alex Madzikova**[4], **Leena S. Chagla**[5], **Jerome Pereira**[6‡], **Mike J. McKirdy**[7‡], **Sankaran Narayanan**[8‡], **Lis Grimsey**[9‡], **Janet Hicks**[5‡], **Ruth James**[10], **Richard M. Rainsbury**[11]

1 Department of Surgery, National University of Science and Technology Medical School, Bulawayo, Zimbabwe, 2 Department of Polymer Technology, National University of Science and Technology, Bulawayo, Zimbabwe, 3 Department of Statistics, National University of Science and Technology, Bulawayo, Zimbabwe, 4 Graduate School of Business, National University of Science and Technology, Bulawayo, Zimbabwe, 5 Burney Breast Unit, St Helens and Knowsley Teaching Hospitals NHS Trust, St Helens, Merseyside, United Kingdom, 6 James Paget University Hospital NHS Foundation Trust, Norfolk, United Kingdom, 7 Clyde Breast Unit, Royal Alexandra Hospital, Paisley, United Kingdom, 8 Breast Unit, University Hospitals of North Midlands, Stoke-on Trent, United Kingdom, 9 Breast Unit, East Sussex Healthcare NHS Trust, Eastbourne, United Kingdom, 10 Luton and Dunstable University Hospital, Luton, United Kingdom, 11 Royal Hampshire County Hospital, Hampshire Hospitals NHS Foundation Trust, Winchester, United Kingdom

☯ These authors contributed equally to this work.
‡ JP, MJM, SN, LG and JH also contributed equally to this work.
* msmagara@hotmail.com

**Data Availability Statement:** All relevant data are within the paper and its Supporting information files.

## Abstract

### Introduction

Breast cancer (BC) is the leading cause of female cancer deaths in Africa, and in Zimbabwe, >80% present with advanced disease. A Needs Project (NP) was carried out to determine the key factors responsible for delayed diagnosis and poor BC outcomes and to investigate possible solutions in 6 rural and urban districts of Matabeleland South and Bulawayo Metropolitan Provinces.

### Methods

A mixed method approach was used to collect data in 2 phases. *Phase 1*: an exploration of key factors leading to poor BC outcomes with >50 professional stakeholders and patient representatives. *Phase 2*: (i) Quantitative arm; validated questionnaires recording breast cancer knowledge, demographic information and perceived barriers to care administered to women and their relatives (Group 1) and health professionals (HPs) (Group 2). (ii) Qualitative arm; 10 focus group discussions with medical specialists and interested lay representatives (Group 3). The Cochran sample size formulae technique was used to determine the quantitative sample size and data was aggregated and analysed using SPSS Version 23™. Purposive sampling for the qualitative study selected participants with an understanding of

**Funding:** RR and LC, on behalf of the Association of Breast Surgery (ABS), applied for a grant from the Tropical Health and Education Trust to fund a Needs Analysis Project in Zimbabwe, to research the bottlenecks in diagnosis and treatment of breast cancer. The Needs Analysis Project was carried out under a memorandum of understanding between ABS and the United Bulawayo Hospitals. This work we are submitting is based on the findings of the the Needs Project. No author received a grant in their individual capacity. Funder- Tropical Health and Education Trust. Grant Award: AGPS 3.04 URL- https://www.THET.org The funder had no role in study design, data collection and analysis, decision to publish, or preparation of the manuscript.

**Competing interests:** The authors have declared that no competing interests exist.

BC and the NP. Focus group discussions were recorded and a thematic analysis of the transcriptions was conducted using NVivo9™.

## Results

Quantitative analysis of Group 1 data (n = 1107) confirmed that younger women (<30years) had the least knowledge of breast cancer (p<0.001). Just under half of all those surveyed regarded breast cancer as incurable. In Group 2 (n = 298) the largest group of health workers represented were general nurses and midwives (74.2%) in keeping with the structure of health provision in Zimbabwe. Analysis confirmed a strong association between age and awareness of BC incidence (p = 0.002) with respondents aged 30–39 years being both the largest group represented and the least knowledgeable, independent of speciality. Nearly all respondents (90%) supported decentralisation of appropriate breast surgical services to provincial and district hospitals backed up by specialist training. Thematic analysis of focus group discussions (Group 3) identified the following as important contributors to late BC diagnosis and poor outcomes: (i) presentation is delayed by poorly educated women and their families who fear BC and high treatment costs (ii) referral is delayed by health professionals with no access to training, skills or diagnostic equipment (iii), treatment is delayed by a disorganised, over-centralized patient pathway, and a lack of specialist care and inter-disciplinary communication

## Conclusion

This study confirms that the reasons for poor BC outcomes in Zimbabwe are complex and multi-factorial. All stakeholders support better user and provider education, diagnostic service reconfiguration, targeted funding, and specialist training.

## Introduction

Breast cancer (BC) is the second most common cause of cancer-related deaths after cervical cancer in Zimbabwean women today [1]. With the increasingly effective detection and treatment of cervical cancer, BC is now the leading cause of cancer-related deaths in all African women [2]. Unfortunately, there is a lack of high-quality cancer data right across Africa and the figures recorded by the Cancer Registries are likely to be just the tip of the iceberg [3]. BC is a growing health problem in Sub-Saharan Africa (SSA), with current rates predicted to double between 2030–2050 [3]. Although BC incidence is relatively low (35/100,000 women in SSA *versus* 120/100,000 women in Europe and North America [4, 5]), 5yr survival rates are three times lower than those enjoyed by women living in high income countries [6]. In the decade 2020–2029, it has been estimated that 416,000 will die from BC in SSA, but one third of these deaths could be prevented by earlier diagnosis and better treatment [7].

Thirty years ago, more than 75% of new BC cases in Zimbabwe presented with either stage 3 or 4 disease [8]; three decades later, the picture has hardly changed [1, 9]. Zimbabwe's National Cancer Registry statistics do not provide a comprehensive picture, as data is collected from urban centres, rather than from the rural locations where 68% of the population live [1, 10]. Typically, patients make slow progress through a lengthy referral pathway, from rural clinics to rural, district and then provincial hospitals, before reaching the central hospitals in Bulawayo or Harare where specialist care is provided [11]. Doctors are resident only at district,

provincial and central hospitals, while municipal clinics, rural clinics and rural hospitals are staffed by health practitioners (HPs).

The Ministry of Health and Child Care (MoHCC) has reported significant gaps in BC management in Zimbabwe [12], with a lack of human resources, treatment facilities and information systems. An understaffed, poorly trained workforce, coupled with remote, high-cost specialist centres combine to make treatment delayed and unaffordable for most of Zimbabwe's deprived population. These factors are most acute in Bulawayo and Matabeleland South (MS), which report the worst BC survival rates in SSA (relative 3-year survival Bulawayo *versus* Namibia, 21.6% *versus* 84.5% [9]. Moreover, screening services for BC (mammography and ultrasound) are available only in private institutions but the cost is prohibitive for the majority. Even among those who can afford screening, there have been insufficient awareness campaigns to encourage people to be screened for cancers [12].

The project's long-term goal is to improve the lives of women in Zimbabwe by raising breast awareness, driving earlier diagnosis, quicker referral and faster access to skilled specialist multidisciplinary care. This Needs Assessment Project (NP) investigated the key factors delaying the diagnosis and treatment of women with suspected BC in Matabeleland South (MS) and Bulawayo Metropolitan Provinces. The primary objective was to highlight the most important and potentially correctable factors causing delay, and to investigate opportunities to address them.

## Materials and methods

### NP: Location, demographics and healthcare provision

The NP was carried out in both MS and the Bulawayo Metropolitan Provinces. The geographical spread of these settings was considered sufficiently representative to identify the bottlenecks in a woman's pathway from rural or urban presentation to referral, diagnosis, and specialist treatment.

Healthcare is provided largely by the state, with patients paying for some tests, drugs and nominal user fees. A patient's access to specialist care is painfully slow, rural roads and transport are unpredictable, distances are long, and accommodation and treatment are unaffordable, so many women seek traditional faith healers instead. HPs (nurses or midwives) run national cervical screening 'VIAC' Visual Imaging with Acetic acid and Cervicography) clinics in 6 hospitals in MS Province (1 Provincial and 5 District Hospitals), all equipped for cervical image transfer, with the potential for breast examination and image transfer. Currently, a paucity of knowledge, skills and training about BC delays referral by 3–6 months, with most patients consulting several HPs before a diagnosis is made. Mammography is available in Bulawayo, but the costs involved make the examination unaffordable for most of the rural and urban populations. Mpilo Central Hospital in Bulawayo is the only facility providing Radiotherapy, Oncology and Histopathology services for the whole region [12].

### Design of the NP

The NP was carried out in three phases:

**Phase 1.** A joint planning group (JPG) representing the Association of Breast Surgery of Great Britain and Ireland (ABS) and United Bulawayo Hospitals (UBH) set up a 3-day meeting in October 2019. More than 50 key professional stakeholders representing healthcare, undergraduate training, academia, medical associations, the ministry of health, public health, the media, patients and patient advocates were invited to attend. The letter of invitation described the meeting as an enquiry into the causes of delayed presentation, diagnosis and treatment of breast cancer with a focus on the referral systems feeding into Bulawayo Central Hospitals.

Attendees were invited to discuss any factors which they thought may act as barriers to earlier diagnosis and treatment, and to put forward practical suggestions for change. The meeting was chaired by one of the authors representing the ABS while two co-authors representing UBH acted as discussion moderators. The meeting was initially planned as a series of separate discussions with individual stakeholder groups. However, several groups were interested in hearing the views of other participants resulting in larger mixed groups where a broader exchange of views occurred. This was accommodated as it was held to reflect the complex and interrelated nature of stakeholder groups in real world practice. Discussions were minuted by the secretary for the UBH board. After each session the discussion was summarised by the chair, giving the opportunity for confirmation or revision by those attending. The three co-authors directly involved in the meeting individually analysed the minutes and jointly reached agreement on emerging themes.

**Phase 2.** This phase focused on exploring the views and needs of the users and providers of healthcare living in MS and Bulawayo Metropolitan Provinces (~1.3 million people).

Questionnaires and interview schedules exploring the key emergent themes from phase 1 were developed, standardized, and validated. Closed question questionnaires designed for quantitative analysis were used to record demographic factors, breast cancer knowledge and practical barriers to accessing health care reported by women and their relatives (Group 1) and health professionals (Group 2). Questionnaires were administered by trained volunteers familiar with the local dialects during June, July and August 2020.

Interview schedules using closed and open questions were employed to structure discussion in 10 focus groups involving medical specialists and lay representatives (Group 3). Two UBH representatives from the JPG and two research fellows acted as moderators and discussions were recorded. Each group was designed to include 6–15 participants. Focus groups were held in each of the 6 MS Province districts and 4 in Bulawayo Metropolitan Province between June and August 2020. The moderators transcribed all video and audio data on to Microsoft Word™ documents. The accuracy of the transcripts was verified with participants before submission for qualitative data analysis.

Phase 2 design originally included an intention to collect hospital level data from patient records as a means of establishing the current baseline for patient pathways. A small pilot project was commenced at Mpilo Hospital (the largest teaching hospital in Bulawayo) to determine feasibility and the most appropriate tools for data collection. This aspect of the project was abandoned due to the poor state of medical records.

**Phase 3.** Data collection was completed, returned, and analyzed by September 2020

## Participants

**Phase 1.** The JPG had a planning session in September 2019 to identify target participants for the phase 1 meeting. It was determined that representatives of the following stakeholder groups were required to provide a comprehensive, contextualised discussion of the range of factors impacting on delayed BC diagnosis in Bulawayo Metropolitan and MS Provinces:

1. Medical speciality groups directly involved in delivering referral and treatment pathways.

2. Patient advocates and breast cancer survivors as actual users of healthcare pathways.

3. Hospital managers responsible for the design and implementation of health care systems.

4. Professional associations with a role in medical training and defining the target skills of members.

5. Political representatives with a role in developing and administering health policy.

6. Media and community leaders with an impact on educating and informing the public.

7. Representatives of health insurance companies.

Members of the JPG representing UBH were able to provide a list of individuals and organizations representative of the target participants. Letters of invitation were issued with a very high acceptance rate resulting in over 50 participants at the phase 1 meeting held in October 2019.

**Phase 2.** *Group 1*: *Women and their families*. Women and their families were recruited from patients attending outpatient clinics at (i) Bulawayo Metropolitan centres including the two central hospitals and the Bulawayo City Council Clinics and (ii) District Hospitals in MS Province including Plumtree (serving Bulilima and Mangwe administrative districts), Esigodini (Umzingwane), Filabusi (Insiza), Maphisa (Matobo), Beit Bridge; and at Gwanda Provincial Hospital.

**Sample size calculation**: The Cochran Sample Size Formulae calculated the required sample size as 1100 respondents from a total population of 1 300 000 people.

*Group 2*: *Health professionals*. Health professionals were recruited from 6 municipal clinics in Bulawayo Metropolitan Province and the 6 district hospitals in MS Province listed above. Municipal clinics are primary health facilities staffed by nurses and midwives whereas district hospitals have an average of 3 resident doctors and radiology, pharmacy, and laboratory staff in addition to nurses. HPs who took part in the quantitative study could not take part in focus group discussions.

**Sample size calculation**: Bulawayo Metropolitan and MS Provinces have an estimated population of 3500 HPs. The Cochran Sample Size Formulae technique determined a sample size of 400 respondents.

*Group 3*: *Medical specialists and community leaders*. Purposive sampling was used to select HPs thought to be knowledgeable about BC and reasons for the NP. Some of the community representatives were identified during the Phase 1 meeting in October 2019 whilst others were selected through a 'snowballing' technique to identify as many advocates as possible within the timeframe and cost constraints. Recruitment was continued until the target focus group size of 6–15 was achieved in each locality. Inclusion criteria for lay members included a minimum level of education that permitted comprehension of the NP objectives (Ordinary Level qualifications). People with no formal education were excluded from the qualitative component of the study.

**Training of data collectors.** Training for trainers and data collectors was conducted in two steps prior to the data collection process.

1. Step 1 involved training trainers in the Bulawayo Metropolitan Province who in turn, trained data collectors in outlying areas. Trainers included 2 surgeons, 2 nurses and 2 research fellows. On completing training, a pilot exercise at a Bulawayo Council Clinic assessed competency in the use of the questionnaires and the internal validity of the instruments. The instruments underwent further revision based on the pilot exercise before being approved for implementation by the JPG.

2. Step 2 involved the selection of hospital staff to collect data on the advice of the local Medical Officers. Each of the 6 districts selected 2 data collectors, and there were 2 data collectors for all the 6 Bulawayo City Clinics (14 in total). Trained staff then carried out the data collection under the supervision of the two UBH representatives of the JPG and two research fellows.

**Data storage.** Data was posted to the Project Office at intervals throughout the project and was then collated and analysed by statisticians. The completed quantitative questionnaires and electronic data were safely stored in lockable steel cabinets. They were subsequently

transferred onto computerized forms for statistical analysis and could only be accessed on password protected devices.

Written consent was obtained from participants in all cases. For face-to-face focus group discussions and interviews, hard copy Consent Forms were distributed allowing participants the time to read through. The Principal Investigator or Core Investigator explained the purpose of the study, the voluntary nature of participation, the right to decline or withdraw at any time, and both confidentiality and anonymity. Volunteer participants signed the consent forms before handing them back to the Core Investigator. Other participants consented online, following the process used for hard copies before signing and posting them back to the Project Office.

## Research design

The study used a mixed method approach, using both qualitative and quantitative data collection instruments.

- **Quantitative data** was collected by trained data collectors from (i) volunteer women (patients presenting with a variety of conditions, including, but not limited to breast diseases), husbands and family members visiting clinics and hospitals, and (ii) from HPs. The questionnaires were designed to provide demographic information, test levels of BC knowledge in the community and in health care providers and identify perceived barriers to health care in a large statistically meaningful sample of the target population. This approach was adopted as a means of triangulating the data acquired from the identification of emergent themes in phase 1 and the qualitative data collected in phase 2. It also provides specific information for targeting recommendations for educational campaigns and redesign of referral pathways.

- **Qualitative data** One of the main objectives of this study was to establish the often complex reasons for a delayed BC diagnosis. It was felt that these reasons were best obtained through a dialogue exploring personal experiences, hence the need for qualitative data. This data was collected by two surgeons and two research fellows during Focus Group Discussions (FGDs) (i) in Matabeleland South Province Districts and (ii) in the Bulawayo Metropolitan area. The findings were then analysed separately and compared. The FGDs held at the Matabeleland South Province Districts were designed to canvass the views of health care professionals and communities about the causes of the late presentation of BC, its diagnosis and treatment. The Bulawayo Metropolitan FGDs were then used to discuss and comment on the views of the district participants, engaging with a wider range of stakeholders including medical specialists, hospital administrators, medical insurance companies, herbalists, faith healers, advocates, and others.

## Data analysis

Quantitative data collected from patients and their relatives, and from HPs were aggregated from outreach areas in the MS Province districts as well as Bulawayo Metropolitan; and an SPSS Version 23™ package used for the analysis. Summary statistics including mean, and mode were calculated for quantitative data and tests for associations were carried out. Qualitative data were transcribed and analysed by NVivo 9™ application, and theoretical triangulation, participant validation and member checks were carried out to ensure the quality of the collected data. Themes of qualitative data analysis and quantitative data were symmetric, supporting the basis for evaluations and buttressed concurrent triangulation. A statistician who holds a master's degree conducted the data analysis.

# Results

## Quantitative arm

**Return rates for questionnaires.**    *Group 1.* A total of 1250 questionnaires were distributed to women (patients) and their relatives in the MS Province districts and Bulawayo Metropolitan Province. Returns numbered 1107 respondents (return rate 88.56%).

*Group 2.* A total of 414 questionnaires were administered; of those 298 were completed and returned (return rate of 71.98%).

## Questionnaire responses

**Group 1: Responses by patients and relatives.**    Data was gathered from 1107 patients and relatives using the structured questionnaires in the MS Province districts and Bulawayo Metropolitan Province. This data was analysed and is presented below for each of the responses.

*(i). Demographics of respondents.* The demographics of respondents are summarized in Table 1.

Overall, the response rate from each region was similar (MS *versus* BM, 55.2% *versus* 43.1%), 82.7% of questionnaires were returned by women, and by a younger demographic (64.1% <50yr). The majority of respondents (64%) reported achieving only a basic level of education at best, and more men achieved Diplomas or Degrees than women- 33.3% versus 24.9%, respectively (see Annex Tables (i) and (ii) in S1 File 'Number of respondents per district' and 'Gender versus education level').

**Table 1.  Demographic characteristics of the respondents.**

|  | Variable | Number | Respondents (%) |
|---|---|---|---|
| **Place of residence** | MS[1] | 611 | 55.2 |
|  | BM[2] | 477 | 43.1 |
|  | NI[3] | 19 | 1.7 |
| **Gender** | male | 184 | 16.6 |
|  | female | 915 | 82.7 |
|  | NI[3] | 8 | 0.7 |
| **Age (yr)** | <30 | 360 | 32.5 |
|  | 30–39 | 350 | 31.6 |
|  | 40–49 | 227 | 20.5 |
|  | >50 | 150 | 13.6 |
|  | NI[3] | 20 | 1.8 |
| **Education level achieved** | Degree | 111 | 10.0 |
|  | Diploma | 176 | 15.9 |
|  | Advanced | 94 | 8.5 |
|  | Ordinary | 502 | 45.3 |
|  | Grade 7 | 167 | 15.1 |
|  | NIL[4] | 40 | 3.6 |
|  | NI[3] | 17 | 1.6 |

[1]MS = Matabeleland South Districts including Beitbridge, Bulilima, Gwanda, Insiza, Mangwe, Matobo.

[2]BM = Bulawayo Metropolitan.

[3]NI = Not Indicated.

[4]NIL = No formal education

*(ii). Knowledge of BC.* Knowledge of BC was found to be statistically dependent on the gender of the respondent, females being more knowledgeable than males (p<0.05). Results of statistical analysis in Table 2 indicate that the null hypothesis of independence between gender and knowledge of BC is rejected.

The strength of the association between gender and knowledge of BC was low since Phi and Cramer's V tests were 0.079 (p = 0.03). Knowledge of BC was related to the age of an individual (p<0.01), with respondents aged 30–49 yr being the most aware of what breast cancer lumps were.

More than 1000 participants responded to questions about their understanding of BC and breast lumps. Nearly two thirds (63.6%) of the whole group declared at least some understanding, those between 30–49 yr were most aware, but the youngest respondents (<30 yr) were the least well informed (p<0.01) (Annex Table (iv) in S1 File 'Knowledge of breast lumps by age group'). Averaging returns from all districts, 65.5%, 29.8% and 4.7% of participants defined BC as a growth, an infection, or 'other condition', respectively. BC was regarded as a problem and something to be feared by 87% and 73% of respondents, respectively. Other conditions thought to present as a breast lump included 'a mole', black magic, 'a mouse', a 'tumour', and a spiritual disease. (Annex Tables (v) and (vi) in S1 File: 'Knowledge of breast lumps by district'; and 'Other definitions of breast cancer by respondents').

*(iii) Family involvement in BC diagnosis.* Most respondents reported they would tell a relative about a BC diagnosis- (54.5% and 26.5% telling spouses and relatives, respectively), and most expected to receive support from family members (71%,).

*(iv) Treatment of BC.* The large majority of >1000 respondents (84.7%) indicated their preference for the treatment of BC in a hospital, with little variation between age groups (range 79.4%-87.2%). Less than 10% expressed their preference for alternative approaches, with only a few (6.1%) choosing treatment in a clinic (Table 3). Just over half (56%) were aware that BC could be cured, and less than a third of those who had developed BC (29.7%) had received treatment. About a third were aware that the disease was treated by chemotherapy and surgery (36.5% and 30.4%, respectively).

*(v) Cancer experiences.* Three quarters of respondents (76.3%) had no experience of BC, either personally or involving a family member. Of those with personal experience, most (85%) had to visit 5 or more doctors or nurses before reaching a diagnosis, and nearly all these women (95.5%) had a further wait of at least 3 months before starting treatment. The top reasons given for this delay included a poor understanding of BC, perceived shortage of medicines and equipment at the hospitals, poverty, and poor access to treatment centres, with very few (<10%) citing cultural or religious beliefs (Annex Table (vii) in S1 File: 'Reasons for delays'). Costs of BC treatment were >USD $625 at the time of the study. Although many poor patients had to rely on personal savings, relatives, and subscription-based medical aid for

**Table 2. BC awareness.**

|  |  | Yes | No | Don't know | Total | p-value |
|---|---|---|---|---|---|---|
| **Gender** | Male | 103 | 62 | 18 | 183 | 0.033 |
|  | Female | 587 | 260 | 53 | 900 |  |
|  | Total | 690 | 322 | 71 | 1083 |  |
| **Age** | Below 30 years | 201 | 114 | 37 | 352 | 0.002 |
|  | 30–39 years | 235 | 95 | 17 | 347 |  |
|  | 40–50 years | 149 | 62 | 13 | 224 |  |
|  | Above 50 years | 100 | 46 | 3 | 149 |  |
|  | Total | 690 | 322 | 71 | 1072 |  |

**Table 3. Preferred provider of treatment by age group.**

| Preference | <30yr | 30-39yr | 40-49yr | ≥50yr | Total |
|---|---|---|---|---|---|
| Clinic | 19 | 20 | 19 | 5 | 63 |
| Hospital | 298 | 287 | 170 | 119 | 874 |
| Faith healer | 12 | 9 | 9 | 4 | 34 |
| Traditional healer | 14 | 12 | 16 | 14 | 56 |
| Other | 1 | 1 | 0 | 2 | 4 |
| Total | 344 | 329 | 214 | 144 | 1031 |
| Hospital (%) | 86.6 | 87.2 | 79.4 | 82.6 | 84.7 |

financial support, only a minority (13.9%) cited high cost as a reason for avoiding clinics and hospitals for treatment.

Exploring access to treatment, three quarters (74.2%) of all respondents reported travelling >5 Km to reach the nearest clinic or hospital. Of 453/1107 responding to this question, over half (57.1%) travelled on foot, 28.6% by bus or car, and the remainder used bike, animal-drawn cart or their other 'own transport' (Annex Table (viii) in S1 File: 'Modes of transport to hospitals or clinics').

**Group 2: Responses by HPs.** Quantitative data was collected from 298 HPs both in MS Districts and Bulawayo Metropolitan Province, using a pretested questionnaire.

*(i) Demographics of HPs.* Younger women accounted for the majority of respondents-68.7% percent were <39 years with only 12.8% >50 years. Most were either general nurses or midwives (74.2%), and nearly half (46.6%) of all respondents were aged 30–39 years (Table 4). Many had lengthy professional experience, with just under half of all HPs (45.2%) reporting >10 years in post (46.5% from Bulawayo Metropolitan Province, compared with 43.2% from MS districts (Annex Tables (ix) and (x) in S1 File: 'Specialisation of health professionals by age group' and 'Work experience of health professionals by district').

*(ii) Awareness of BC and knowledge or support for the project.* Just over half of HPs (54.5%) [159/292] were aware of the incidence of BC (Annex Table (xi) in S1 File 'Awareness of BC incidence by age group'). There was strong evidence of an association between age and awareness of BC incidence (p = 0.002) at the 5% level of significance, with respondents aged 30–39 years being the least knowledgeable age group (Annex Table (xi) in S1 File), but awareness was independent of specialization (p>0.05). Half of the respondents were also unaware that the Ministry of Health and Child Care's national database captured details of breast examination and advice given to patients with abnormalities (Table 5). Two (0.8%) and 4 (1.3%) respondents did not indicate their experience and specialty, respectively.

Knowledge about the Bulawayo Breast Cancer Project was low (33.67% of all respondents), and variable (0% in Beitbridge and Bulilima Districts to 73.6% in Gwanda), but 90% supported the project once they got to know the project's objectives. (Annex Table (xii) in S1 File: 'Health professionals' knowledge and support for the breast cancer project').

*(iii) Training, surgical services and access to histopathology.* Nearly all health practitioners (97.3%) supported greater cancer specialization, and most (93%) were willing to undergo relevant training if this was made available. Most also agreed that training in US and core biopsy would improve BC diagnosis, and 94% were willing to learn these skills. Two thirds expressed concern about the shortage of trained radiographers, and one third agreed that nurses running the cervical screening (VIAC) clinics should be considered for BC training programmes.

Nearly all respondents (90%) supported decentralization of appropriate breast surgical services to provincial and district hospitals (Fig 1), backed up by specialist training.

**Table 4. HPs Responses by demographic profile, experience and specialty.**

|  | Variable | Number | Respondents (%) |
|---|---|---|---|
| **Gender** | Male | 87 | 29.2 |
|  | Female | 211 | 70.8 |
| **Age (yr)** | <30 | 66 | 22.1 |
|  | 30–39 | 139 | 46.6 |
|  | 40–49 | 55 | 18.5 |
|  | ≥50 | 38 | 12.8 |
| **Experience (yr)** | <5 | 94 | 31.5 |
|  | 5–10 | 68 | 22.8 |
|  | 11–20 | 91 | 30.5 |
|  | 21–30 | 23 | 7.7 |
|  | >30 | 20 | 6.7 |
| **Specialty** | Nurse | 158 | 53.0 |
|  | Midwife | 63 | 21.1 |
|  | Surgery | 12 | 4.0 |
|  | Radiography | 12 | 4.0 |
|  | Pathology | 3 | 1.0 |
|  | Radiology | 2 | 0.7 |
|  | Oncology | 1 | 0.3 |
|  | Other | 43 | 14.4 |

Practitioners in all districts expressed concern about the ability of current public histopathology services to deal with the increased number of specimens generated by the project. Nearly half (46.0%) thought the number of public laboratories was insufficient, while 44.3% did not know/did not have an opinion (Annex Table (xiii) in S1 File: 'Availability of histopathology laboratories'), Eighty four percent of respondents agreed that the turnover time of breast biopsies would greatly improve if a governmental or non-governmental funding model could be secured, similar to the voucher system developed for the cervical screening programme.

**Table 5. Responses by health professionals.**

|  | Variable | Number | [1]Respondents Knowledgeable (%) |
|---|---|---|---|
| **Awareness of breast cancer** | <30 yr | 42 | 64.6 |
|  | 30–39 yr | 59 | 43.0 |
|  | 40–49 yr | 31 | 58.4 |
|  | ≥50 yr | 27 | 72.9 |
| **Knowledge of the project** | Yes | 100 | 33.6 |
|  | No | 156 | 52.4 |
|  | Don't know | 32 | 10.7 |
| **Support for the project** | Yes | 249 | 83.6 |
|  | No | 21 | 7.0 |
|  | Don't know | 17 | 5.7 |

[1]Proportion of respondents in each age group

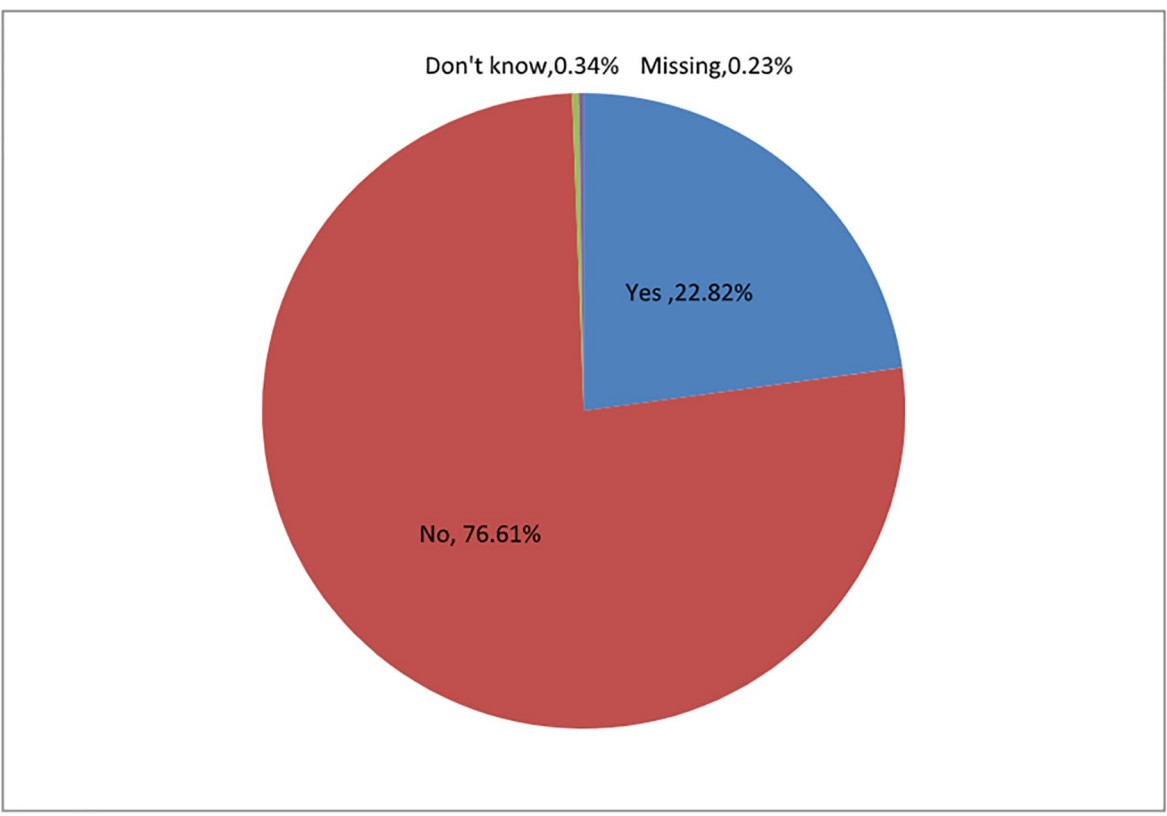

**Fig 1. Health professionals' responses to restriction of breast surgical services to major referral hospitals.**

### Qualitative arm

A total of 109 participants took part in the 10 focus group discussion (FGD) meetings held in MS and Bulawayo Metropolitan Provinces.

1. FGDs in MS (6 groups): Groups were mixed and consisted of HPs (doctors, nurses, laboratory scientists, pharmacists, radiographers, and other health workers) and local community leaders (councillors, village heads, church leaders). Some groups included complimentary health providers in the form of traditional healers and herbalists. There were no patient advocate groups represented.

2. FGDs in Bulawayo Metropolitan (4 groups): Each group represented a distinct stakeholder group.

   **FGD1.**   Patient advocacy groups, and non-governmental organizations. Half of the women who attended as advocates were BC survivors.
   **FGD2.**   Community leaders, traditional healers, church leaders, herbalists, and school head teachers.
   **FGD3.**   Health services administrators including a hospital CEO and senior pharmacist involved in medicines procurement, healthcare fund managers and a banker involved in loan management for health services.
   **FGD4.**   Medical specialists involved in the care of women including general surgeons, gynaecologists, oncologists, and radiologist.

A thematic analysis was carried out using Nvivo9™. There was a high level of concordance between the groups, with urban areas demonstrating more breast cancer awareness and rural areas placing more emphasis on the importance of family involvement in making health decisions. The major themes are summarized in Table 6.

## Discussion

### Summary of the study and findings

MS Province's low density, deprived population (810,000) returns one of Zimbabwe's lowest scores for human development, income and education, and the highest score for families requiring emergency support [13]; 25% live >10km from the nearest clinic, and 75% can't afford medication or a bus fare [14]. This unique project was set up to investigate why women in MS and Bulawayo Metropolitan Provinces present so late with BC, and experience such poor outcomes in this most deprived environment. The early involvement of professional stakeholders to identify the most likely reasons for these outcomes was a crucial factor informing the design and content of the qualitative and quantitative questionnaires used in data collection.

The study revealed multiple reasons for the parlous outcomes of women with BC in Zimbabwe. High levels of ignorance about the disease were displayed not only by the general public but also by their healthcare providers. This was compounded by women's fears and anxieties, and a concern about the effectiveness and high cost of conventional treatment (PD, patient delay). Finally, delays at every stage of a patient's pathway combine with a lack of

**Table 6. The management of BC: Key messages identified by the FGDs.**

| Themes | [1]MS issues | [2]BM issues | Key messages |
|---|---|---|---|
| **Awareness** | Low everywhere, with differences between districts. Lack of health centre-based educational materials. No BC focus in community health campaigns | More aware than MS, some with experience of BC, diagnosis, treatment and counselling. Late presentation in urban communities requires public education | The lack of BC awareness in the community reflects the historical neglect of education, public health initiatives and national campaigns |
| **Diagnosis** | Delayed by fear, low awareness, beliefs, poor services and equipment. Family permission required | Diagnosis delayed by a lack of basic diagnostic equipment, coupled with unaffordable costs of investigation | Ignorance, fear and cultural factors combine with poor services/equipment to delay diagnosis |
| **Treatment** | Traditional and faith healers trusted more than doctors and nurses until disease became advanced | Women usually decide to go to affordable faith or traditional healers, continuing 'treatment' until disease advances | Traditional is more popular than conventional medicine particularly in rural communities, until BC becomes advanced |
| **Travel** | Greater distances equated with higher costs for travel and accommodation- a particular concern in Matobo District (Maphisa Hospital) | Distance travelled for treatment is a real problem, some patients travelling very long distances, or overseas if affordable | Travelling long distances for treatment is a major problem for many patients |
| **Cost** | Costs of transport, lodgings, food and treatment is a universal problem that could be helped by a media campaign to raise public awareness, and decentralising diagnostic services | Real concerns expressed about high costs of investigation and treatment, often requiring self-funding after exhausting medical insurance | The high costs of conventional BC investigation and treatment are unaffordable for most women living in poor communities |
| **Family involvement** | High value is placed on the involvement of the whole family in decision-making in most districts | The role of partners and families was not raised as an issue | Families in rural communities play a key role in decisions about the location and type of treatment |
| **Patient information** | Good communication would help to convince people that BC is curable, but was totally lacking between doctors, nurses and oncologists | No coordinated large-scale patient information/management systems are available, but piloting of an electronic health database is ongoing | Lack of standardised information and data collection combine with poor interprofessional communication to undermine trust in conventional treatment |

[1]MS- Matabeleland South Province

[2]BM- Bulawayo Metropolitan Province

specialization, communication and data collection to prevent any progress towards earlier detection and treatment (HD, health system delay). Despite these dismal findings, a response rate of >80% to the study questions indicated a high level of interest in BC in the region, particularly by those <50yr who accounted for >80% of all responders. This high return rate and a high level of agreement between all groups also strengthens the validity of the findings.

## Comparison with international studies

The findings of our study are consistent with those reported from a range of low and middle income countries (LMICs) around the world- that a greater number of women who live in remote or rural areas present with BC at a more advanced stage, compared with those living in urban locations [15, 16]. International comparisons also show that these women face a greater delay in the diagnosis and treatment of BC [17], because of poor access to often remote diagnostic facilities [18], and a lack of adequate numbers of trained HPs [19]. Studies of the factors that lead to diagnostic delay in different parts of the world generally focus on different aspects of the diagnostic pathway. In a comparison of 24 studies in LMICs with 29 studies in higher income countries, those in LMICs generally focus on PDs leading to delayed presentation, whilst those in developed countries focus on HDs- such as delays between presentation, diagnosis and treatment, once the patient enters the health system [20]. Our study is unique, as we investigated the role of both PDs and HDs in the late presentation of BC in Zimbabwean women. We have shown a whole system failure that demands whole system solutions.

## Comparison with national studies

Our observations support the findings of a growing number of studies from other SSA countries which have highlighted some common factors leading to a shocking situation where nothing has changed for decades. In keeping with this study, most report very low levels of BC awareness, both in the general population and amongst HPs [21, 22], leading to long delays in presentation [23] and high rates of advanced disease [24, 25] particularly in lower socio-economic groups who often place their trust in traditional healers [26]. A recent review article by Gbenonsi *et al* explored health system factors impacting on diagnostic and treatment pathways in 13 out of the 48 SSA countries [27]. Their findings identified delayed waiting times caused by complex referral pathways, geographical inaccessibility of health care facilities, lack of medical training and high costs, as key barriers to earlier BC diagnosis. Whilst their conclusions indicate similar health system failings to those identified in the current study, the authors did not include any studies relating to Zimbabwean populations, thus demonstrating a lack of published research in this area.

In common with Rwanda [28], we found that women in rural communities including MS are further handicapped by poor access to remote, ill-equipped clinics, which can lead to delays of >6 months [29], with a further impact on the stage at diagnosis [30]. Improved access to local diagnostic and treatment supports attendance and encourages earlier presentation [31]. Although population-based screening is currently unavailable and unaffordable in MS, down-sizing the stage at presentation of BC can be achieved by simple population-level interventions such as the education of women, communities, volunteers and HPs [32], and enjoyed the near-universal support of our respondents.

## Strengths of the study

The strength of this study is its uniqueness in being conducted simultaneously in both rural and urban settings, while including a full range of users, providers and managers of the regional health service. This enabled a wide-ranging and in-depth perspective of the needs and problems

experienced, not only by BC patients, but also the whole public health system. Gbenonsi *et al* recommend further primary studies looking at the scope for adapting health policies for local contexts [27]. This study addresses these issues comprehensively, with a view to directly informing future health strategies in MS and the Bulawayo Metropolitan Provinces. Gbenonsi *et al* acknowledge that a weakness in the data underpinning their review was that most studies involved the experience of women with BC who have reached hospital level care, thus limiting the representativeness and generalisability of the data. By contrast, our quantitative analysis included women where a personal history of BC was unknown at the point of entry into the study. This gave the potential to acquire data on women who had a wider range of health system experience.

Information gathered during phase 1 not only generated emergent themes, but resulted in a rich contextual background which was used to interpret and triangulate the findings in the quantitative and qualitative arms of the study. The study design enabled the collection of quantitative information from a meaningful, statistically calculated sample size of the target population. The data collected is of great relevance to the targeting of future educational initiatives amongst the general population and health care providers in Bulawayo and MS Provinces. The qualitative data provides a further analysis of the complex reality behind delayed diagnosis from the point of view of health care providers and community members. The key findings were concordant with the data derived from phase 1 and the quantitative study, but the situated nature of the qualitative arm in which focus groups were held, and reflected the specific experience of local communities, revealed particular regional and local concerns which are important for future health care planning.

## Key messages

All quantitative and qualitative data from responders- patients and relatives, HPs and focus groups- has been analysed and grouped together to generate a number of 'key messages' below. These have been developed to inform the implementation of interventions which challenge the *status quo* and have potential to bring about significant change.

- *Poor understanding about BC*: knowledge about BC is extremely limited, both in the general population and amongst HPs- 74% of whom are nurses or midwives. Although nearly all respondents recognize BC as a problem and something to fear, the two most poorly informed groups are young women (<30yr), and HPs aged 30-39yr- the largest and most experienced group of nurses and midwives responsible for caregiving, advice and referral. This problem is most acute in MS, and reflects years of neglected education, poor public health initiatives and community programmes, leading to little change in Zimbabwe's understanding of BC over the last decade [33].

- *Fear and anxiety*: heightened by a lack of any personal or family experience of BC, little or no access to basic diagnostic equipment and long waits for referral and treatment.

- *Lack of trust*: suspicion of conventional medicine is compounded by longstanding family and community trust in traditional healers and witchcraft, postponing any referral for conventional investigation until too late for curative treatment in many cases, particularly in less educated rural communities.

- *Cost*: many patients cannot afford the expenses incurred by the long distances travelled, accommodation, costly investigation and treatment, all of which quickly deplete any personal savings or modest medical insurance cover.

- *Delays*: related to a complex, lengthy and hierarchical referral pathways, a lack of local diagnostic clinics and expertise, and a severe shortage of trained specialist staff, diagnostic

equipment, and histopathology services in the two major referral centres. These delays have been overcome by introducing an NGO-funded voucher system for the 'VIAC' programme that reimburses the patient's costs of specimen transport, and examination in a private pathology laboratory.

- *Failure to support specialization*: three quarters of all HPs are generalists without any specialist training or experience. More than 90% support specialist training in the diagnosis and management of BC, combined with decentralization of initial diagnostic services to provincial and district hospitals.

- *Lack of data collection and poor communication*: paper-based clinic and hospital records are often poor or absent, and there is no access to electronic records or patient information systems. Combined with the failure of communication between central hospitals, referring clinics and communities, this parlous situation further undermines patient confidence and trust in an already fragmented treatment pathway.

## Potential strategies for policymakers

Much can be done to address these problems, including workforce training at all levels, community and public awareness initiatives, service reorganisation, and financial investment in equipment and information technology (IT). This will involve a number of coordinated developments that policymakers will need to consider to achieve change:

- *Address the widespread ignorance about BC* by training doctors, health practitioners, and volunteers, using 'training the trainer' principles to cascade learning and assessment. A mixture of blended and interactive online learning approaches will increase accessibility and penetration in more remote settings.

- *Overcome the fear, anxiety and a lack of trust in conventional medicine* through a programme of community and public engagement, facilitated by trained volunteers working with rural communities, clinic HPs, advocacy groups, the MoHCC and other agencies.

- *Minimize the long delays and the high costs of BC treatment* by transforming the current referral pathway to provide new diagnostic BC clinics in MS, overseen by the Provincial Medical Director. When run by trained staff using diagnostic ultrasound equipment, IT and online reporting, they will avoid costly and distressing onward referral for most women. Project partners will work with NGOs, the MoHCC and others to offset high treatment costs with a voucher system.

- *Embed specialist practice, data collection and audit into the BC service* by appointing fellows to UK breast units to acquire breast diagnostic, communication and audit skills. These skills can be cascaded throughout the workforce, backed up by hands-on and online training opportunities. Working alongside the Government's Non-Communicable Disease (NCD) and Health Informatic initiatives will support the development of systems to monitor and audit BC outcomes to track changes in detection and downstaging.

## Limitations of the study

The Covid -19 pandemic was the greatest limitation of the physical implementation the NP. Travel was restricted, gatherings discouraged, and many Government regulations were in place. Permission to travel had to be sought from UBH, and was generally granted, allowing successful completion of data collection. Factors that may have influenced responses and

reduced the reliability of the study included language or dialect problems limiting the understanding of questions, ignorance about the nature of BC and suspicion of 'western medicine', concerns about costs of diagnosis and treatment, and the potential for volunteers to assist personally with the completion of questionnaire responses to meet study deadlines. Participants in Group 1 (women and family members) may not have been representative of the local communities, as they already had health issues or other reasons for attending the local clinics and hospitals. It is acknowledged that the 6 MS focus group discussions may have been impacted by their combined structure in which health specialists and lay members debated the study issues together. This has the potential to limit the freedom for lay members to engage because of a sense of hierarchy in which professional opinions may be held to be more valid. In practice, these groups demonstrated a high degree of concordance in their views which correlated with the views expressed by the women in the FGD 1 in Bulawayo Metropolitan Province, women and their families in the quantitative arm of the study, and with patient advocates in phase 1.

## Conclusions

Women living in Matabeleland South and Bulawayo Metropolitan Provinces experience myriad factors that delay the diagnosis of BC and face the direst outcomes seen anywhere in SSA. This study interrogated >1500 users, providers, and leaders of healthcare in the region, and found unanimous agreement about the key barriers preventing the timely diagnosis and treatment of BC. These barriers obstruct and frustrate a patient's journey through the system at every level. First, presentation is delayed by women and their families who are poorly informed and fearful about BC, and the high costs of treatment. Second, referral is delayed by HPs who have no access to training, skills or diagnostic equipment. Lastly, treatment is delayed by a fragmented, over-centralised patient pathway, lack of specialization and poor communication at all levels of the public health service. Understanding and breaking down these barriers is the first step in a coordinated programme to bring about change.

## Supporting information

**S1 File.**
(DOCX)

## Acknowledgments

Lucy Davies- coordinated communication among the authors.

Narcisius Dzvanga, Rudo Chikodzore, Richard Sithole and Primrose Chipandu–provided logistical support for data collection.

## Author Contributions

**Conceptualization:** Munyaradzi S. Magara, Simbarashe G. Mungazi, Peeps Gonde, Alex Madzikova, Leena S. Chagla, Jerome Pereira, Mike J. McKirdy, Sankaran Narayanan, Lis Grimsey, Janet Hicks, Richard M. Rainsbury.

**Data curation:** Munyaradzi S. Magara, Simbarashe G. Mungazi, Peeps Gonde, Hausitoe Nare, Desmond Mwembe, Alex Madzikova, Leena S. Chagla, Ruth James, Richard M. Rainsbury.

**Formal analysis:** Hausitoe Nare, Desmond Mwembe, Ruth James.

**Funding acquisition:** Leena S. Chagla, Richard M. Rainsbury.

**Investigation:** Munyaradzi S. Magara, Simbarashe G. Mungazi, Peeps Gonde, Alex Madzikova, Richard M. Rainsbury.

**Methodology:** Munyaradzi S. Magara, Simbarashe G. Mungazi, Peeps Gonde, Hausitoe Nare, Desmond Mwembe, Alex Madzikova, Leena S. Chagla, Jerome Pereira, Mike J. McKirdy, Sankaran Narayanan, Lis Grimsey, Janet Hicks, Ruth James, Richard M. Rainsbury.

**Project administration:** Munyaradzi S. Magara, Richard M. Rainsbury.

**Resources:** Munyaradzi S. Magara, Simbarashe G. Mungazi, Peeps Gonde, Alex Madzikova, Leena S. Chagla, Richard M. Rainsbury.

**Software:** Hausitoe Nare, Desmond Mwembe, Alex Madzikova.

**Supervision:** Munyaradzi S. Magara, Richard M. Rainsbury.

**Validation:** Peeps Gonde, Hausitoe Nare, Desmond Mwembe, Leena S. Chagla, Ruth James, Richard M. Rainsbury.

**Visualization:** Munyaradzi S. Magara, Peeps Gonde, Hausitoe Nare, Desmond Mwembe, Alex Madzikova.

**Writing – original draft:** Munyaradzi S. Magara, Simbarashe G. Mungazi, Peeps Gonde, Hausitoe Nare, Desmond Mwembe, Alex Madzikova, Leena S. Chagla, Richard M. Rainsbury.

**Writing – review & editing:** Munyaradzi S. Magara, Simbarashe G. Mungazi, Peeps Gonde, Alex Madzikova, Leena S. Chagla, Jerome Pereira, Mike J. McKirdy, Sankaran Narayanan, Lis Grimsey, Janet Hicks, Ruth James, Richard M. Rainsbury.

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
