## [Decision Letter · Decision Letter 0]

2 Dec 2021

PONE-D-21-13224Factors leading to the late diagnosis and poor outcomes of breast cancer in Matabeleland South and Bulawayo Metropolitan Provinces in ZimbabwePLOS ONE

Dear Dr. Magara,

Thank you for submitting your manuscript to PLOS ONE. After careful consideration, we feel that it has merit but does not fully meet PLOS ONE’s publication criteria as it currently stands. Therefore, we invite you to submit a revised version of the manuscript that addresses the points raised during the review process. Please understand that the peer review process was longer than expected, including a re-assignment step in the academic editor role in early October. The presented research in the manuscript is highly relevant and the Authors are encouraged and invited to submit the revised version of their manuscript along the reviewer suggestions. Besides, please supplement the manuscript with reference on the Cochran's Sample Size Formula and show or describe the raw data belonging to Table 5 (now only the statistical test output is presented, without the underlying data). Please also consider native speaker review to improve the language of the manuscript.   

We look forward to receiving your revised manuscript.

Kind regards,

János G. Pitter, MD, PhD

Academic Editor

PLOS ONE

Journal Requirements:

2.  Please include the full name of all ethics committees that approved your study in the manuscript Methods.

3. Please provide additional details regarding participant consent. In the ethics statement in the Methods and online submission information, please ensure that you have specified what type you obtained (for instance, written or verbal, and if verbal, how it was documented and witnessed). If your study included minors, state whether you obtained consent from parents or guardians."

4. In your Methods section, please provide additional information about the participant recruitment method and the demographic details of your participants. Please ensure you have provided sufficient details to replicate the analyses such as: a) a detailed description of any inclusion/exclusion criteria that were applied to participant recruitment, c) b) a description of how participants were recruited."

5. To comply with PLOS ONE submission guidelines, in your Methods section, please provide additional information regarding your statistical analyses. For more information on PLOS ONE' expectations for statistical reporting, please see " ext-link-type="uri" xlink:type="simple">https://journals.plos.org/plosone/s/submission-guidelines.#loc-statistical-reporting."

6. Please ensure you have discussed any potential limitations of your study in the Discussion, including study design, sample size and/or potential confounders.

7. We note that Figure 1 in your submission contain [map/satellite] images which may be copyrighted. All PLOS content is published under the Creative Commons Attribution License (CC BY 4.0), which means that the manuscript, images, and Supporting Information files will be freely available online, and any third party is permitted to access, download, copy, distribute, and use these materials in any way, even commercially, with proper attribution. For these reasons, we cannot publish previously copyrighted maps or satellite images created using proprietary data, such as Google software (Google Maps, Street View, and Earth). For more information, see our copyright guidelines: http://journals.plos.org/plosone/s/licenses-and-copyright.

Natural Earth (public domain): http://www.naturalearthdata.com/ blic domain): http://eol.jsc.nasa.gov/sseop/clickmap/ Maps at the CIA (public domain): https://www.cia.gov/library/publications/the-world-factbook/index.html and https://www.cia.gov/library/publications/cia-maps-publications/index.html NASA Earth Observatory (public domain): http://earthobservatory.nasa.gov/ Landsat: http://landsat.visibleearth.nasa.gov/ USGS EROS (Earth Resources Observatory and Science (EROS) Center) (public domain): http://eros.usgs.gov/# Natural Earth (public domain): http://www.naturalearthdata.com/

Reviewers' comments:

Reviewer's Responses to Questions

**Comments to the Author**

1. Is the manuscript technically sound, and do the data support the conclusions?

Reviewer #1: No

Reviewer #2: Partly

2. Has the statistical analysis been performed appropriately and rigorously? 

Reviewer #1: No

Reviewer #2: No

3. Have the authors made all data underlying the findings in their manuscript fully available?

Reviewer #1: Yes

Reviewer #2: Yes

4. Is the manuscript presented in an intelligible fashion and written in standard English?

Reviewer #1: No

Reviewer #2: No

5. Review Comments to the Author

Reviewer #1: Although this paper addresses an interesting issue on Factors leading to late diagnosis and poor outcomes of breast cancer in Matabeleland South and Bulawayo Metropolitan Provinces in Zimbabwe, It needs major /critical revisions before considered for publication including editorial and grammatical corrections. In addition I recommend the authors to follow the author guidelines of PLOS ONE Journal.

Abstract

- Methods : the authors should clearly define the outcome variable, the method of data analysis, the study design

Background:

- The background section would be more interesting if the authors could expand the section to include more information about the available breast cancer diagnostics and treatment centers in the study hospital and better to compare it with other developing countries in Africa.

- cite the references before the punctuation marks

-

I. Methods

- study area: it will be good if the authors should add more information about the study area (the availability of the diagnostic and treatment centers in the hospital, average patient flow per a specific period of time, the presence or absence of chemotherapy, radiotherapy services , available professionals on the area …

- I suggest the authors to clarify /to show the following points

study populations and sample populations

inclusion and exclusion criteria’s

Where the ethical clearance was obtained?

How the participants are selected or included in the study, what was the sampling technique?

How the sample size was determined?

Data analysis

- On the data analysis section, the authors should briefed about the statistical analysis in detail for quantitative data (descriptive statistics and inferential statistics) and the methods of qualitative data analysis (thematic content or framework analysis ...)

- Who transcribed the qualitative data?

- How the trustworthiness of qualitative data was assured (credibility, dependability, transferability, confirmability?

- It will be good to mention the qualification and experience of those who conducted the qualitative data collection and analysis

- could you please justify the need of qualitative data since the objectives of the study can be addressed in qualitative methods alone

Result

- the tables can be merged and please minimize the number of tables (merge table 1.2.3.4 ) : the tables are not attractive since you prepare a table for a single variable

- when you write your result, please avoid unnecessary associations you made, just strict to answer your objective i.e. factors associated to delay diagnosis of breast cancer and to breast cancer out come only, therefore you can make association with this two outcome variables only.

- The result section is not informative and needs rewriting!! please look other papers conducted on late diagnosis of breast cancer in Africa

- the figures are also need modification, please look journal guideline

- For the qualitative data , please show the themes , categories and codes identified from the participant narrations

II. Discussion and conclusion

- The discussion section needs comparing your results with other studies conducted globally and you have justify the similarities and difference between the studies ; nothing you do in this area

- What are the strengths and limitation of your study?

Reviewer #2: The authors present a mixed methods study on barriers to timely diagnosis of breast cancer in Zimbabwe. The manuscript duly reflects the amount of work that was completed. There are major revisions that could enhance the quality and value of the study. In particular, the manuscript can be reduced 30-50% in length without compromising key findings and a greater emphasis on focused key messages can improve readability and impact.

Major considerations:

(1) Presentation of findings and length: there are many significant findings that are informative. However, a significant proportion of the results can be moved to an Annex allowing for more streamlined, efficient narrative that highlights key points. Emphasis should be made on the results that support the primary conclusions. For example,

a. Knowledge of breast cancer by district and/or work experience by District: what implications do these data have? Can this be deleted? If included, is there are association between the two?

b. Demographics/characteristics of respondents and health professionals can each be summarized in one table (and cross tabulation of age and educational levels can be presented without a table.

Data can also be better presented to improve the efficiency of the content. For example, the FGDs can be consolidated into one major section, focusing on key findings and any relevant differences identified. Content like “Word Clouds” can also be moved to an Annex.

(2) Framework: the authors present a conceptual framework, but there is no explanation for how it was developed and/or reference to any existing cancer early diagnosis framework developed by WHO or other public health agency. Furthermore, the presented framework is not used to help organize the content, leaving the reader without a clear reference to process the material presented. Using an existing framework can further orient the findings around key outcomes (beyond the target on late stage disease) such as delays in specific intervals along diagnostic pathway, treatment access and/or abandonment.

(3) Statistical review: there are several additional analyses that could improve the data quality. For example,

a. Sub-groups: multiple times in the study, a sub-group is presented without clarification as to why the provided data do not sum to 100% or to the group percentage (eg, lines 345-346, 349-350, 356-357)

b. Analyses do not consider confounding factors. For example, the awareness of cancer incidence should be done by age and by profession.

Additionally, clarification should also be provided regarding select results/findings. For example, what is the difference between “No” and “Don’t Know” as it pertains to “Knowledge about breast lumps (Table 6)? Also, how why is inadequate medicine considered a delay when not all breast cancers require medicines?

Minor considerations:

(1) Key findings should be clarified and made more coherent: for example, there are recommendations for greater specialization and also decentralization. How would the authors advise policy-makers to interpret these two findings in the short term through a coherent approach?

(2) Survey questions: the inclusion of “Have you ever discovered a breast lump” and “Did someone ever examine you and find a breast lump?” would be informative to help guide primary care planning. These data are not well known and were valuable to have included in the questionnaire. Furthermore, can further insights be provided on the women who travel abroad to seek care? This is also particularly relevant to understand.

(3) VIAC voucher system: given that this is part of the recommendations, it would be helpful to provide the reader with additional background.

(4) MoH: was consideration given to interview Ministry of Health representatives as a key stakeholder? If not, why not?

(5) Data collectors: how many were trained?

(6) “Patients and relatives”: can an exact breakdown be provided?

(7) Copy edits: the manuscript would benefit from copy editing (eg, 714-718) to aid the reader

(8) Reference: please add reference for sentence 102-104 (and consider exchanging “manpower” with “health workforce”

(9) Breast cancer as witchcraft: this was referenced in the FGD but not reflected to the same extent in the stakeholder interviews (eg, Table 8). Can the authors provide their interpretation of the frequency of such belief systems in the Discussion?

6. PLOS authors have the option to publish the peer review history of their article (what does this mean?). If published, this will include your full peer review and any attached files.

Reviewer #1: No

Reviewer #2: No

---

## [Author Response · Author response to Decision Letter 0]

30 Mar 2022

The reviewer and editor comments have been addressed in the "Response to Reviewers" document, uploaded with other resubmission files. Kindly see our responses to the comments in the attached files section.

---

## [Decision Letter · Decision Letter 1]

15 Jul 2022

PONE-D-21-13224R1

Factors leading to the late diagnosis and poor outcomes of breast cancer in Matabeleland South and Bulawayo Metropolitan Provinces in Zimbabwe

PLOS ONE

Dear Dr. Magara,

Thank you for submitting your manuscript to PLOS ONE. After careful consideration, we feel that it has merit but does not fully meet PLOS ONE’s publication criteria as it currently stands. Therefore, we invite you to submit a revised version of the manuscript that addresses the points raised during the review process.

Thank you for the detailed revisions to your manuscript. We have had your submission re-assessed by one of the original reviewers, who feels that there are still some issues that need addressing (see comments below).

In particular, could you provide further details regarding the methodology of the qualitative component of your research.

Also, the reviewer mentions a recent systematic review/meta-analysis on the topic of breast cancer diagnosis in sub-Saharan Africa. I believe that the review in question may be this one:

Gbenonsi, G., Boucham, M., Belrhiti, Z. et al. Health system factors that influence diagnostic and treatment intervals in women with breast cancer in sub-Saharan Africa: a systematic review. BMC Public Health 21, 1325 (2021). https://doi.org/10.1186/s12889-021-11296-5

I understand that this paper was published after you initially submitted your work to PLOS ONE. However, I agree with the reviewer that a discussion of your findings in relation to those of Gbenosi et al. is relevant.

If applicable, we recommend that you deposit your laboratory protocols in protocols.io to enhance the reproducibility of your results. Protocols.io assigns your protocol its own identifier (DOI) so that it can be cited independently in the future. For instructions see: https://journals.plos.org/plosone/s/submission-guidelines#loc-laboratory-protocols. Additionally, PLOS ONE offers an option for publishing peer-reviewed Lab Protocol articles, which describe protocols hosted on protocols.io. Read more information on sharing protocols at https://plos.org/protocols?utm_medium=editorial-emailutm_source=authorlettersutm_campaign=protocols.

We look forward to receiving your revised manuscript.

Kind regards,

Steve Zimmerman, PhD

Associate Editor, PLOS ONE

Reviewers' comments:

Reviewer's Responses to Questions

**Comments to the Author**

1. If the authors have adequately addressed your comments raised in a previous round of review and you feel that this manuscript is now acceptable for publication, you may indicate that here to bypass the “Comments to the Author” section, enter your conflict of interest statement in the “Confidential to Editor” section, and submit your "Accept" recommendation.

Reviewer #1: (No Response)

2. Is the manuscript technically sound, and do the data support the conclusions?

Reviewer #1: Partly

3. Has the statistical analysis been performed appropriately and rigorously? 

Reviewer #1: No

4. Have the authors made all data underlying the findings in their manuscript fully available?

Reviewer #1: Yes

5. Is the manuscript presented in an intelligible fashion and written in standard English?

Reviewer #1: No

6. Review Comments to the Author

Reviewer #1: General comments

This title of the paper raised an important issue on the factors contributing to late diagnosis and poor outcomes of breast cancer in Zimbabwe. My concern here is the authors mentioned as they used a mixed study approach /design but the abstract section, the result tells us about the qualitative findings, so it should describe about the factors which are associated to the outcome variable i.e late diagnosis and poor outcome (quantitative findings?

The other point is in qualitative research, there are issues which need to be addressed clearly i.e. issue of trustworthiness (dependability, transferability, confirmability,) should be clearly mentioned in the methodology data analysis procedure section?

In addition, in sub-saharan Africa, there is a systematic review and meta-analysis which shows the barriers for early diagnosis of breast cancer, and most of the results of this work are duplicates, so what makes this study original for the area? Since there are studies which tried to assess the factors for late diagnosis and outcome in both qualitatively and quantitatively even there is a review!

I recommend the authors to revise their document based on the author guidelines of the journal!!

7. PLOS authors have the option to publish the peer review history of their article (what does this mean?). If published, this will include your full peer review and any attached files.

Reviewer #1: No

---

## [Author Response · Author response to Decision Letter 1]

14 Sep 2022

Response to Reviewers’ Comments

25 August 2022

Dear Dr Zimmerman,

Factors leading to the late diagnosis and poor outcome of Breast Cancer in 

Matabeleland South and the Bulawayo Metropolitan Provinces in Zimbabwe

PONE-D-21-13224R1

Thank you for your letter regarding our resubmission of the above paper investigating breast cancer outcomes in Zimbabwe. Our responses to your own comments and those of your reviewer are laid out below:

Response to Dr Zimmerman

1. Further details regarding methodology and the qualitative component of your research:

These are now summarised in the Abstract (Lines 38-89), and comprehensively explained in the rewritten Materials and Methods section (Lines 165-439). Also see our attached responses 1 and 2 to Reviewer #1.

2. The reviewer mentions a recent systematic review/meta-analysis on the topic of breast cancer diagnosis in sub-Saharan Africa. I understand that this paper was published after you initially submitted your work to PLOS ONE. However, I agree with the reviewer that a discussion of your findings in relation to those of Gbenonsi et al. is relevant:

This publication has now been cited in the Discussion section (Line 1191-1200). A comparison with our study has highlighted how we have addressed several of the recommendations proposed by the systematic review, including (i) the need to adapt health policies to the local context- this was the reason for focusing specifically on Matabeleland South and Bulawayo, and (ii) the need for more generalisable and representative data- again our study focused on a cross-section of women and their families in the general population, rather than those already being treated for breast cancer. Also see our attached response 3 to Reviewer #1.

Response to Reviewer #1

We have summarised our responses to the Reviewer’s three main concerns in the Table below. New text addressing these concerns is tracked in the abstract, methods and discussion sections of the resubmitted paper: 

Reviewer’s general comments

1. ‘…the authors mentioned as they used a mixed study approach /design but the abstract section, the result tells us about the qualitative findings, so it should describe about the factors which are associated to the outcome variable i.e late diagnosis and poor outcome (quantitative findings)?’

 1. The abstract section has been revised to include the key quantitative findings for each group. (Lines 38-89)

2. ‘…in qualitative research, there are issues which need to be addressed clearly i.e. issues of trustworthiness should be clearly mentioned in the methodology data analysis procedure section…’ 2. The methods section has been revised substantially to include further information on recruitment, the composition of each study group, study conduct and the recording and analysis of material. (Lines 165-439). As now clarified in this section, the trustworthiness of the data has been ensured in four ways, following a set of 4 criteria originally proposed by Lincoln and Guba (1985) including: (i) ‘Credibility: the extent to which the researchers interpretation is endorsed by those with whom it was conducted’. The draft paper was widely circulated to the Joint Planning Group of stakeholders including patients and patient advocacy groups for their input and comments before finalising the submission.

(ii) ‘Transferability: the researcher should provide enough rich detail for the reader to be able to assess whether the conclusions drawn in one setting can transfer to another’. By including a wide, representative sample of health service users and providers, both during the stakeholders meeting, and during deployment of the research questionnaires in a variety of locations, we ensured the transferability of the findings to other similar demographic groups and similar healthcare settings.

(iii) ‘Trackable variance: the researcher should demonstrate that they have taken into account the inherent instability of the phenomenon they are studying’. This has been addressed in the ‘strengths and limitations of the study’ subsection in the discussion section

(iv) ‘Confirmability: there should be sufficient detail of the process of data collection and analysis so that the reader can see how the researchers arrived at their conclusions’. The phases and process of data collection are now addressed in greater detail in the methods section, and a thorough review of the accuracy, relevance and meaning of the raw data by several independent members of the Project Board not involved in the design or delivery of the field study was undertaken, greatly strengthening confirmability.

The methods section also includes a new section on the justification for the methodology used (Lines 447-489).

3. ‘In addition, in sub-saharan Africa, there is a systematic review and meta-analysis which shows the barriers for early diagnosis of breast cancer, and most of the results of this work are duplicates, so what makes this study original for the area? Since there are studies which tried to assess the factors for late diagnosis and outcome in both qualitatively and quantitatively even there is a review…’ The systematic review by Gbenonsi et al referred to by your Reviewer includes studies of health systems in 13 out of 48 SSA countries, none of which involve Zimbabwe. The authors have emphasised the need for further studies exploring health needs as they relate to local contexts. The current study explores heath needs in a context specific manner which result in practical suggestions for change. In this sense the Gbenonsi article supports the publication of this article. The discussion section has been revised to include the Gbenonsi systematic review (Lines 1191-1200) and the strengths and limitations of the study section has been extended (Lines 1234-1280).

Yours sincerely

Munyaradzi Magara

---

## [Decision Letter · Decision Letter 2]

12 Dec 2022

PONE-D-21-13224R2Factors leading to the late diagnosis and poor outcomes of breast cancer in Matabeleland South and Bulawayo Metropolitan Provinces in ZimbabwePLOS ONE

Dear Dr. Samson Magara,

Thank you for submitting your manuscript to PLOS ONE. After careful consideration, we feel that it has merit but does not fully meet PLOS ONE’s publication criteria as it currently stands. Therefore, we invite you to submit a revised version of the manuscript that addresses the points raised during the review process.

**Comments to the Author**

1. If the authors have adequately addressed your comments raised in a previous round of review and you feel that this manuscript is now acceptable for publication, you may indicate that here to bypass the “Comments to the Author” section, enter your conflict of interest statement in the “Confidential to Editor” section, and submit your "Accept" recommendation.

Reviewer #1: All comments have been addressed

Reviewer #3: (No Response)

2. Is the manuscript technically sound, and do the data support the conclusions?

Reviewer #1: Yes

Reviewer #3: Partly

3. Has the statistical analysis been performed appropriately and rigorously?

Reviewer #1: Yes

Reviewer #3: Yes

4. Have the authors made all data underlying the findings in their manuscript fully available?

Reviewer #1: Yes

Reviewer #3: Yes

5. Is the manuscript presented in an intelligible fashion and written in standard English?

Reviewer #1: Yes

Reviewer #3: No

6. Review Comments to the Author

Reviewer #1: The manuscript is potential for publications . The authors addressed my comments in the revised document

Reviewer #3: 1. The framing of sentences in the manuscript is poor which needs to be rephrased.

2. The materials and methods have to descriptive at the same time it should be on point and precise. For example- The population details of MS and Bulawayo Metropolitan provinces is not necessary in the methodology section.

3. Statement regarding ethical approval has to come in the end section of materials and methodologies.

4. The various steps conducted in the study methodology process can be written in precise manner, highlighting the important points.

5. The sub-headings and contents of the materials and methods section needs reframing. For example- Justification for Methodology can be written as Research Design and Qualitative data section has been poorly framed.

6. The legends of the tables also needs to be re-consider. For example- Table 1- Demographic characteristics of the respondents.

7. Table 2 is not giving a clear description of the significant association between the variables used for Breast Cancer awareness. Hence, restructuring of the table need to be done incorporating the factors used for the significant test.

8. Discussion part has been poorly framed.

First part of discussion should discuss the brief details of the study and its prime findings.

Second part of discussion should compare the findings with the international studies done under similar objectives with rationale.

Third part of discussion should compare the findings with the national studies done under similar objectives with rationale

Fourth part of discussion should be the superficial inference drawn for this study citing the strengths

9. Key messages mentioned in this manuscript should be after discussion part.

10. The sequence should be-

1. Discussion

2. Key messages

3. "Advice to policy- makers implementing changes" (Change it to - Potential strategies for policy-makers)

4. Limitations of the study

5. Conclusions

6. Acknowledgements

7. References

7. PLOS authors have the option to publish the peer review history of their article (what does this mean?). If published, this will include your full peer review and any attached files.

**Do you want your identity to be public for this peer review?** For information about this choice, including consent withdrawal, please see our Privacy Policy.

Reviewer #1: **Yes: **Aragaw Tesfaw

Reviewer #3: No

If applicable, we recommend that you deposit your laboratory protocols in protocols.io to enhance the reproducibility of your results. Protocols.io assigns your protocol its own identifier (DOI) so that it can be cited independently in the future. For instructions see: https://journals.plos.org/plosone/s/submission-guidelines#loc-laboratory-protocols. Additionally, PLOS ONE offers an option for publishing peer-reviewed Lab Protocol articles, which describe protocols hosted on protocols.io. Read more information on sharing protocols at https://plos.org/protocols?utm_medium=editorial-emailutm_source=authorlettersutm_campaign=protocols.

We look forward to receiving your revised manuscript.

Kind regards,

Rajiv Janardhanan, Ph.D.

Academic Editor

PLOS ONE
---

## [Author Response · Author response to Decision Letter 2]

31 Jan 2023

Response to Reviewers’ Comments

1. If the authors have adequately addressed your comments raised in a previous round of review and you feel that this manuscript is now acceptable for publication, you may indicate that here to bypass the “Comments to the Author” section, enter your conflict of interest statement in the “Confidential to Editor” section, and submit your "Accept" recommendation.

Reviewer #1: All comments have been addressed

Reviewer #3: (No Response)

2. Is the manuscript technically sound, and do the data support the conclusions?

Reviewer #1: Yes

Reviewer #3: Partly

3. Has the statistical analysis been performed appropriately and rigorously?

Reviewer #1: Yes

Reviewer #3: Yes

4. Have the authors made all data underlying the findings in their manuscript fully available?

Reviewer #1: Yes

Reviewer #3: Yes

5. Is the manuscript presented in an intelligible fashion and written in standard English?

Reviewer #1: Yes

Reviewer #3: No

6. Review Comments to the Author

Reviewer #1: The manuscript is potential for publications . The authors addressed my comments in the revised document

Reviewer #3: 

1. The framing of sentences in the manuscript is poor which needs to be rephrased. 

It's difficult to address this comment without knowing which sentences/paragraphs your Reviewer is referring to, short of rephrasing/rewriting the whole document. This issue was not raised in relation to either of the 2 previous submissions, or indeed by any of your previous reviewers. The senior author carried out all final edits, including this third iteration, and has published extensively without encountering any comments of this kind. We trust that any concerns of this nature have now been addressed satisfactorily in the sections below that have been revised in line with Reviewer 3’s suggestions. 

2. The materials and methods must descriptive at the same time it should be on point and precise. For example- The population details of MS and Bulawayo Metropolitan provinces is not necessary in the methodology section.

The population details have been moved to the opening paragraph of the discussion (Lines 581-586). 

3. Statement regarding ethical approval has to come in the end section of materials and methodologies.

This statement has been moved to the end of the materials and methods section under a new subtitle ‘Ethical approval and consent’ (Lines 288-300)

4. The various steps conducted in the study methodology process can be written in precise manner, highlighting the important points.

See point 5 below.

5. The sub-headings and contents of the materials and methods section needs reframing. For example- Justification for Methodology can be written as Research Design and Qualitative data section has been poorly framed.

The subtitle ‘Justification for Methodology’ has been replaced by ‘Research Design’ (Line 303), and the ‘Qualitative data’ subsection has been rewritten to clarify the rationale and methods used during the qualitative data collection phase (Lines 317-333).

6. The legends of the tables also needs to be re-consider. For example- Table 1- Demographic characteristics of the respondents.

Legends of Tables 1, 3 and 4 changed as suggested.

7. Table 2 is not giving a clear description of the significant association between the variables used for Breast Cancer awareness. Hence, restructuring of the table need to be done incorporating the factors used for the significant test.

Table 2 and associated text have been revised as suggested, now incorporating the parameters for tests of association. (Lines 404 – 414).

8. Discussion part has been poorly framed.

First part of discussion should discuss the brief details of the study and its prime findings.

See new subsection of Discussion ‘Summary of the study and findings’ (Lines 579-602).

Second part of discussion should compare the findings with the international studies done under similar objectives with rationale.

See new subsection of Discussion ‘Comparison with international studies’, with 6 additional references (Lines 604-619).

Third part of discussion should compare the findings with the national studies done under similar objectives with rationale

See new subsection of Discussion ‘Comparison with national studies’, with one additional reference (Lines 623-646).

Fourth part of discussion should be the superficial inference drawn for this study citing the strengths.

See new subsection of Discussion ‘Strengths of the study’ (Lines 648-674)

9. Key messages mentioned in this manuscript should be after discussion part.

74See new position of ‘Key messages’ (Lines 680-774).

10. The sequence should be-

1. Discussion

See relocated ‘Discussion’ section (Lines 578—674).

2. Key messages

See new position of ‘Key messages’ (Lines 680-731).

3. "Advice to policy- makers implementing changes" (Change it to - Potential strategies for policy-makers)

See new position ‘Potential strategies for policy-makers’ (Lines 776-801).

4. Limitations of the study

See new position for ‘Limitations of study’ (Lines 803-848).

5. Conclusions

See new position for ‘Conclusions’ (Lines 877-889).

6. Acknowledgements

See new position for ‘Acknowledgements’ (Lines 896-899).

7. References

See new position for ‘References’, with seven additional references relating to (Lines 901-1001)

---

## [Decision Letter · Decision Letter 3]

14 Sep 2023

Factors leading to the late diagnosis and poor outcomes of breast cancer in Matabeleland South and Bulawayo Metropolitan Provinces in Zimbabwe

PONE-D-21-13224R3

Dear Dr. Magara

We’re pleased to inform you that your manuscript has been judged scientifically suitable for publication and will be formally accepted for publication once it meets all outstanding technical requirements.

Kind regards,

Nontuthuzelo Iris Muriel Somdyala, Ph.D

Academic Editor

PLOS ONE

Additional Editor Comments (optional):

Dear Author

This manuscript has been in circulation of peer review for quite a long time. I understand it has been revised three times. However, I am keen to decide on this manuscript. Please send me the original and revised copies of your manuscript so as make an informed decision.

Thank you so much for your understanding and patience.

Dr Somdyala - Academic Editor

Reviewers' comments:

Reviewer's Responses to Questions

**Comments to the Author**

1. If the authors have adequately addressed your comments raised in a previous round of review and you feel that this manuscript is now acceptable for publication, you may indicate that here to bypass the “Comments to the Author” section, enter your conflict of interest statement in the “Confidential to Editor” section, and submit your "Accept" recommendation.

Reviewer #3: All comments have been addressed

2. Is the manuscript technically sound, and do the data support the conclusions?

Reviewer #3: Yes

3. Has the statistical analysis been performed appropriately and rigorously? 

Reviewer #3: Yes

4. Have the authors made all data underlying the findings in their manuscript fully available?

Reviewer #3: Yes

5. Is the manuscript presented in an intelligible fashion and written in standard English?

Reviewer #3: No

6. Review Comments to the Author

Reviewer #3: Thank you for addressing all the comments and making the recommended changes in the manuscript. However, some minor revisions needs to be done. Kindly find the attached manuscript and see the comment section.

7. PLOS authors have the option to publish the peer review history of their article (what does this mean?). If published, this will include your full peer review and any attached files.

Reviewer #3: No

---

## [Editor Report · Acceptance letter]

25 Oct 2023

PONE-D-21-13224R3 

Factors leading to the late diagnosis and poor outcomes of breast cancer in Matabeleland South and the Bulawayo Metropolitan Provinces in Zimbabwe 

Dear Dr. Magara:

I'm pleased to inform you that your manuscript has been deemed suitable for publication in PLOS ONE. Congratulations! Your manuscript is now with our production department. 

Kind regards, 

on behalf of

Dr. Nontuthuzelo Iris Muriel Somdyala 

Academic Editor

PLOS ONE